# Pollen Production of Selected Grass Species in Russia and India at the Levels of Anther, Flower and Inflorescence

**DOI:** 10.3390/plants11030285

**Published:** 2022-01-21

**Authors:** Elena Severova, Yury Kopylov-Guskov, Yulia Selezneva, Vera Karaseva, Shrirang R. Yadav, Dmitry Sokoloff

**Affiliations:** 1Department of Higher Plants, Biological Faculty, M.V. Lomonosov Moscow State University, 119234 Moscow, Russia; yurez-kg@yandex.ru (Y.K.-G.); sokoloff-V@yandex.ru (D.S.); 2Faculty of Biology, Shenzhen MSU-BIT University, Shenzhen 518172, China; 3Institute of Natural Science, S.A. Esenin Ryazan State University, 390000 Ryazan, Russia; posevina_julia@mail.ru (Y.S.); v.karaseva@365.rsu.edu.ru (V.K.); 4Angiosperm Taxonomy Laboratory, Department of Botany, Shivaji University, Kolhapur 416004, India; sryadavdu@rediffmail.com

**Keywords:** pollen, Poaceae, pollen/ovule ratio, reproductive biology, Europe, tropical Asia, nrITS

## Abstract

Grasses produce large amounts of pollen and are among the main causes of pollen allergy worldwide. Quantification of the roles of individual grass species in airborne pollen is an important task, because morphologically indistinguishable pollen grains of different species may differ in allergenicity. This requires knowledge of the pollen production of individual grass species; however, accumulated data are insufficient in this respect. Attempting to fill this gap, we studied pollen production per inflorescence in 29 grass species which are widespread in Middle Russia and India. Pollen production per inflorescence is determined by the number of grains per anther, the number of flowers in a spikelet and the number of spikelets per inflorescence, with the latter parameter being the most variable. We support the hypothesis that pollen production per inflorescence differs significantly between annual and perennial grasses. The greater pollen production of perennials can be interpreted as a tendency to guarantee cross-fertilization of species with self-incompatibility. The inferred pollen/ovule (P/O) ratios suggest the occurrence of facultative xenogamy in all annuals and obligate xenogamy in most perennials in the present dataset, though some self-incompatible annuals exist outside our sampling. Earlier data indicated that the P/O ratio of the annual cereal crop rye (*Secale cereale*) is higher than in any annual or perennial species sampled here. A ratio of pollen production to seed set (P/S ratio) is suggested to be another efficient parameter in reproductive biology of grasses. We highlight a need for detailed studies of reproductive biology in grasses that include both pollen and seed production. We found a correlation between pollen production per anther and anther length. A rough approximation of c. 1000 pollen grains per 1 mm of the length of an anther provides an instrument for estimates of pollen production in plant communities.

## 1. Introduction

Grasses (Poaceae) make up the second-largest monocot family, comprising more than 11,000 species worldwide [1]. The importance of this group lies not only in the number of species and their wide distribution; interactions between grasses and humans are multidimensional, and go far beyond their use as crop cereals [2,3] and pasture plants [4]. Grasses produce a large amount of pollen, and are among the main causes of pollen allergies [5,6]. To date, eleven groups of grass pollen allergens have been identified; groups 1 and 5 are responsible for most (up to 95%) cases of the grass allergy [7].

The very high pollen production characteristic of many grass species is related to their reproductive strategy. Most grasses are anemophilous [8,9] with the exceptions of some cleistogamous and a few entomophilous species [10]. Cleistogamy was reported in some tropical as well as temperate grasses, including several varieties of the most important cereal crop, wheat [11,12,13]. For a number of species, mainly from the tropics, environmental control of cleistogamy was found [14,15,16,17]. Entomophily is documented or suspected to occur in some tropical grasses, as in the absence of air movement under the canopy of a tropical forest, anemophily is inefficient [14]. Soderstrom and Calderon [18] described pollination by flies and gall midges of species of *Parianum* and *Olyra* in tropical South America. Pollen of *Parianum* has a well-sculpted exine [18], which agrees with the idea of the occurrence of entomophily in this taxon. Various insects were observed as visitors of anthetic inflorescences of other tropical grasses [19,20,21,22,23,24,25,26], but their participation in effective pollination was not always obvious. Besides reproductive strategies and local conditions, pollen production -of grass species depends on the life form (annual or perennial). Perennial grasses are known to produce more pollen than annuals [27,28,29].

Pollen production of grass species has been estimated previously as a part of research on pollination biology [30,31,32,33,34], seed production [35], the breeding and selection of domesticated cereal grasses [36,37,38,39] and aerobiology [27,28,40,41,42]. Pollen production per anther can differ significantly between species, as well as among different populations of the same species. A good (and almost the only) example of pollen production estimation of a species occurring in multiple climatic and geographical conditions is available for *Dactylis glomerata*. Its pollen production can differ significantly in different regions, i.e., from 1300–1800 pollen grains (pg)/anther [40,42] to 3000–3500 pg/anther [27,32].

Investigations of grass pollen production are in great demand in aerobiology. Airborne pollen monitoring is of a primary importance for the interpretation of allergy data and therapy planning, but also for agronomy, environmental health, forestry etc. [5]. Poaceae is a stenopalynous family. It is impossible to distinguish among the pollen grains of different grass genera, let alone species, using light microscopy [43]. Pollen curves and calendars obtained by routine aerobiological monitoring normally identify grasses as one group. The determination of the precise flowering times of different species and their pollen concentrations may be very important for allergy sufferers as, despite the lack of detailed studies, it is assumed that different grass species may have pollen grains with different degrees of allergenicity [44]. Refining data on grass pollination period down to the species level is possible either on the basis of phenological observations [29] or by metabarcoding [45,46,47,48]. Both approaches need data on the pollen production of different species to determine the impact of individual species and validate the results.

Another factor closely related to investigations of pollen production is potential correlations between the number of pollen grains per anther, anther length and pollen size. Grasses (and sedges) produce pollen in an unusual manner. The microspores and then pollen grains form a single layer around the secretory tapetum, so that any single pollen grain has a direct contact between its distal aperture and tapetal cells [49,50,51]. In general, such a location restricts the number of pollen grains per locule. Friedman and Harder [52] discovered a positive correlation between pollen size and the number of pollen grains per anther and total volume of pollen per flower. They hypothesized that “species with large pollen invest more in pollen than species with small pollen. (…) This greater investment may signal compensation for lower pollination efficiency (proportion of removed pollen grains exported to stigmas)” [52]. However, this hypothesis needs to be tested. Revealing correlations between pollen size and anther length may be important from an agricultural point of view, i.e., in the search for good pollen donors for hybridization. Such study was carried out for wheat varieties [39], revealing positive correlations between these parameters. Detailed studies of variation in pollen yield in five species of *Bromus* revealed that anther length is an excellent predictor of pollen production in this genus [34]. A link between anther length and pollen production was found in *Triticum*, *Secale* and their hybrid *Triticale* [36,37,38,39].

Despite the importance of data on the pollen production of individual species of grasses, they are still fragmentary. Most studies performed so far have been concentrated on grasses in particular geographical regions. It is therefore difficult to say to what extent their conclusions can be generalized and used elsewhere. The present study is intended to contribute another dataset regarding the pollen production of grasses, and one that would make it possible to perform large-scale comparisons.

To investigate the pollen production of grasses with different life forms and possibly different reproductive strategies, we selected two regions with contrasting climatic conditions: Central Russia (temperate continental climate) and the Western Ghats region near Kolhapur, Maharashtra (India) (tropical monsoon climate) [53]. In Central Russia, we chose two sampling areas, Moscow oblast and Ryazan oblast within the same climatic zone [53], to evaluate the variability of pollen production within a biogeographical region. The pollen production of the grasses of Maharashtra, Moscow oblast and Ryazan oblast has never been studied before. There are 415 species of grasses recorded in Maharashtra [54]. The flora of Central Russia includes 270 species of grasses [55]. The most complete study of grass pollen production in India was carried out by Subba Reddy and Reddy [31] in Andhra Pradesh, and covered 54 species. The state of Andhra Pradesh is situated at the eastern part of peninsular India and includes a considerable part of Eastern Ghats. Investigations of grass pollen production in Central Russia are even more incomplete than in India. We found the only study with data on the pollen production of eleven grass species from Saratov oblast [32]. The aim of our study was (1) to evaluate pollen production per inflorescence in common grass species in Maharashtra and Middle Russia [54,55]; (2) to compare the pollen production of species with different life forms; (3) to evaluate the variability of all traits that determine pollen production; and (4) to determine whether pollen production is correlated with the size of pollen grains and anther length in a large-scale analysis that includes various grass species that differ in ecology and reproductive biology.

## 2. Material and Methods

### 2.1. Sampling Areas

The plants were collected in Kolhapur (16°41′ N, 74°14′ E) and surrounding parts of the Maharashtra state (India), Moscow (55°45′ N, 37°37′ E) and Ryazan (54°36′ N, 39°42′ E) and their surroundings (Russia) from sites comprising different grass habitats, mainly in cities and suburbs, i.e., recreation areas (parks, botanical gardens), lawns and wastelands. These habitats were selected as we assume their primary roles in the production of allergic pollen affecting the local population.

### 2.2. Species Identification

The species were identified using local floras [54,55]. The taxonomy of a number of tropical grass groups is still far from being stable due to the lack of global revisions utilizing extensive molecular phylogenetic data. Anticipating possible future taxonomic adjustments, in addition to traditional herbarium vouchers, we generated nrITS sequences of all accessions of Indian grass species studied here and deposited them in GenBank. These can be used as molecular vouchers in any potential complex taxonomic situations. We extracted DNA from dried leaves using DiamondDNA Plant Kit (“AltaiBioTech” Ltd., Barnaul, Russia) according to the manufacturer’s protocol. The quality and concentration of the extracted DNA were evaluated with a spectrophotometer. We amplified the nuclear complete ITS region with ITS4 and ITS5 primers [56]. The reaction mix (20 μL) contained 10–30 ng of template DNA, 5 pmol of each primer and MasDDTaqMIX ready-to-use mix: (200 μM of each dNTP, 2.0 mM MgC12, 1.5 unit of Taq-polymerase, buffer solution; “Dialat” Ltd., Moscow, Russia). PCR conditions were as follows: 5 min at 95 °C—preliminary denaturation; 30 cycles of: 30 s at 95 °C—denaturation, 30 s at 54 °C—primer annealing, 40 s at 72 °C—elongation; 7 min. at 72 °C—final elongation. Amplicons were purified with CleanUp Mini Kit (“Evrogen” Ltd., Moscow, Russia). Commercial service of Sanger sequencing was provided by the “Evrogen” Ltd. (Moscow, Russia). All obtained sequences we manually checked by comparing with chromatograms in the MEGA X software [57].

A total of 15 species which are widespread in Middle Russia and 14 Indian species were studied. All studied species with voucher information are listed in Table 1. Voucher specimens were deposited in the Herbarium of Moscow University (MW).

### 2.3. Estimation of Pollen Production

The method for pollen production estimation was chosen according to the objectives of our research. When studying the breeding systems and reproductive strategies of particular species, pollen production per anther or per flower is usually calculated [31,32]. For aerobiological monitoring, it is important to estimate pollen production of each species in a certain territory; as such, different methods of abundance estimation are used [27,28,42]. In our study, we estimated pollen production per inflorescence. Our approach allowed us to use the results in both areas, namely, to investigate the reproductive strategies of species and to evaluate pollen production per territory, as the number of inflorescences and their densities in various types of vegetation can be relatively easily counted in the field. To estimate pollen production per inflorescence, we investigated several parameters. The mean number of spikelets was calculated based on 20 inflorescences for Russian species and 10 for Indian species. To determine the number of flowers per inflorescence, we selected three spikelets from each inflorescence. The number of flowers per inflorescence was obtained by multiplying the average number of flowers/spikelets by the average number of spikelets/inflorescence. To calculate the total number of pollen grains per anther, a mature pre-anthetic flower from each inflorescence was selected. The number of pollen grains was estimated using a method of Kaybeleva and Yudakova [32]. Each anther was crushed on a glass slide to create a pollen monolayer in a drop of water, which was then studied and photographed under light microscope Nikon Eclipse Ci (Nikon GMBH, Dusseldorf, Germany). Pollen was counted using the Adobe Photoshop software [58]. *Arrhenatherum elatius* has dimorphic fertile flowers in a spikelet, with one flower being male and the other bisexual. As studies of another grass with dimorphic flowers, *Hierochloe*, revealed functional differences between the anthers of female-fertile and female-sterile flowers, at least in some species [59], we calculated pollen production per anther independently for these flower types in *Arrhenatherum*. Total pollen production per inflorescence was calculated by multiplying the number of pollen grains per anther by the number of anthers per flower by the number of flowers per inflorescence. We determined a variance of total pollen production per inflorescence according to a formula for variance of product of random variables. To estimate the average anther length, 10–20 anthers from different inflorescences were measured with a Nikon Eclipse Ci microscope (Japan). The exact sample sizes for each species are shown on Figure 1. The anther length was defined as the length of its longer theca when the two thecae were of unequal length. The biggest diameter of pollen grains was measured on temporary slides in water under a light microscope. At least 20 nonacetolyzed pollen grains of each species were studied. We used the Student’s t-test and Tukey’s HSD test to make pairwise and multiple comparisons, respectively, and linear coefficients of correlation to describe connections among variables (Pearson’s r for parametric and Spearman’s rho for nonparametric data). Data analyses were performed in R 4.0.5 [60].

## 3. Results

### 3.1. ITS1-2 Sequencing

For all 14 species from India examined here, the ITS1-2 region was successfully amplified and sequenced. The sequences deposed in GenBank were the result of combining two raw sequences (derived from forward and reverse primers). For five species (*Digitaria stricta*, *Isachne elegans*, *Iseilema hackelii*, *Setaria intermedia*, *Urochloa panicoides*), these were the first ITS1-2 sequences in the GenBank database. For the remaining nine species, such sequences were already present in the database and showed a high level of identity (>98% for seven species; 91.2% and 93.5% for *Tripogon jacquemontii* and *Urochloa ramosa*, respectively) with our newly contributed data.

### 3.2. Number of Pollen Grains per Anther

The mean number of pollen grains per anther varied between 106 in *Digitaria stricta* (India) and 5495 in *Arrhenatherum elatius* (Russia, Moscow) (Table 2, Figure 1). Differences between species of the same genus were revealed in *Poa*, *Digitaria*, *Chloris* and *Urochloa*. In the cases of *Chloris* (*C. barbata* and *C. virgata*), *Digitaria* (*D. ciliaris*, *D. stricta* and *D. sanguinalis*) and *Urochloa* (*U. ramosa* and *U. panicoides*), pollen production differed with high levels of significance (Tukey’s HSD test; *p*-values << 0.001). Differences between the three studied species of *Poa* (*P. annua*, *P. pratensis* and *P. trivialis*) were less significant (Tukey’s HSD test; *p*-values < 0.01). Pollen production per anther was almost the same in *Setaria intermedia* and *S. pumila*, despite the fact that studied samples were collected in different climatic zones. *Lolium perenne* and *Festuca pratensis*, two species traditionally classified in different genera, but currently placed in *Lolium* sensu lato [9], were close in pollen production per anther. Pollen production per anther of all of the annual species sampled here was significantly lower compared with that of perennial species (*t*-test; *p*-value << 0.001). For the perennial species (*Calamagrostis epigeios*, *Dactylis glomerata*, *Elymus repens*, *Lolium perenne*, *Phleum pratense*, *Poa trivialis*) sampled in two locations within the same climatic zone (Moscow and Ryazan), no significant difference in pollen production per anther was observed between the two localities (Tukey’s HSD test; *p*-values > 0.05; Figure 2). We also found no difference in pollen production between male and bisexual flowers of *Arrhenatherum elatius* (*t*-test; *p*-value > 0.05); therefore, in subsequent analyses, we used our measurements of this species as a single dataset.

### 3.3. Pollen Production per Inflorescence

The number of fertile flowers per spikelet was constant in 17 species, i.e., one or (*Arrhenatherum elatius*) two. Among other species, the mean number of flowers per spikelet varied from 2.1 (*Poa trivialis*) to 9 (*Tripogon jacquemontii*). This parameter was rather stable, with a coefficient of variation not exceeding 35% (*Tripogon jacquemontii*). Most annuals examined here had spikelets with one flower; the exceptions were *Eleusine indica* and *Poa annua*, with the latter species being only a facultatively annual plant. The number of spikelets per inflorescence, in contrast, was the most variable parameter. The mean values ranged from 16 (*Elymus repens,* Ryazan) to 2638 (*Calamagrostis epigeios*, Moscow), with the coefficient of variation ranging from 11–14% (*Lolium perenne*) to 52–57% (*Phleum pratense*). A wide range of variability was typical for both annual and perennial species.

Pollen production per inflorescence varied between genera, between species within the same genus and between different samples of a species (Figure 3). The means ranged from c. 48,000 for *Setaria pumila* to c. 9,500,000 for *Calamagrostis epigeios*. The pollen production of the obligately annual species sampled here was significantly lower compared with that of perennial species (*t*-test; *p*-value << 0.001; *Poa annua* was not included in this analysis, as it cannot be precisely identified as either annual or perennial species) and varied from c. 40,000 (*Urochloa ramosa*) to c. 617,000 (*Eleusine indica*). Pollen production per inflorescence in the perennial species *Iseilema hackelii* was within the range of variation found among annual species sampled here. The mean pollen production of the other perennials studied here was higher than that in annuals, though the figures found for *Elymus repens* and *Tripogon jacquemontii* approached those of the most productive annual *Eleusine indica*. The pollen production per inflorescence of *Poa annua* fell into the range observed for obligate annual species.

In perennial species, we found a significant but not strong negative correlation between the number of spikelets per inflorescence and pollen production per anther (Spearman’s rho = −0.529; *p*-value < 0.05), i.e., species with a low amount of pollen in their anthers tended to have lots of spikelets in their inflorescences.

Variability of pollen production per inflorescence within a genus was studied for *Chloris*, *Digitaria*, *Setaria* and *Poa*. The most significant differences were in *Poa* between *P. annua* and obligately perennial species. The pollen production of the obligately perennial species was 7–8 times higher than in *P. annua*. *Chloris barbata* and *C. virgata* differed in pollen production by 2.9 times, *Setaria intermedia* and *S. pumila*—by 4.3 times. The pollen production of the two Indian species of *Digitaria* was very similar, and 1.3–1.6 times lower than that of *D. sanguinalis* from Russia.

The pollen production per inflorescence of *Calamagrostis epigeios*, *Dactylis glomerata*, *Phleum pratense*, *Poa trivialis* and *Festuca pratensis* differed between samples: for *Calamagrostis epigeios* and *Poa trivialis*, the values were higher in Moscow, while the others showed higher values in Ryazan (Figure 4; *t*-test, all the *p*-values < 0.05). For *Bromus inermis, Poa annua*, *Elymus repens* and *Lolium perenne*, no significant difference in pollen production per inflorescence between the two Russian locations was observed.

### 3.4. Anther Length, Diameter of Pollen Grains and Pollen Production per Anther

No strong correlation was found between pollen grain size and pollen production per anther in our sample set. A strong and significant positive correlation (Pearson’s r = 0.857; *p*-value << 0.001) was observed between the anther length and the pollen production per anther (Figure 5). Our equation of linear regression was the following: pollen production = 1024.4 × length (the intercept did not significantly differ from 0; the coefficient was significant at *p*-value << 0.001; R^2^ = 0.735). This tendency was clear for both regions (Russia and India) and for plants of both life forms (annual and perennial; r and *p*-values not shown). We also revealed a positive correlation between the diameter of pollen grain and anther length (Pearson’s r = 0.652; *p*-value < 0.01), which means that longer anthers tend to develop larger pollen grains. The pollen grains of annual species usually did not exceed 30 ϻm in diameter, with the exception of *Setaria pumila* and *Digitaria sanguinalis*, although some perennial plants also had pollen grains of this size (*Isachne elegans*, *Iseilema hackelii*, *Tripogon jacquemonti*, *Calamagrostis epigeios*, *Poa trivialis*, *Alopecurus pratensis*). Pollen grains with the biggest diameter belonged to perennial plants (*Bromus*, *Lolium*), but the size of pollen differed considerably among different samples of the same species (in *Bromus*, *Dactylis*, *Lolium*, *Phleum*; Table 2).

### 3.5. Pollen/Ovule Ratio

Grass flowers have no more than one ovule, so when all flowers are uniform and bisexual, the pollen production per flower is equal to P/O ratio, a parameter that shows the ratio between pollen production and the number of ovules. The P/O ratios of all obligate annuals sampled here were lower than those of facultative annuals and perennials (Figure 6). The P/O ratios calculated for the two samples of the facultatively annual species *Poa annua* slightly exceeded those of the obligate annuals (Figure 6). The highest P/O ratio was found in *Arrenatherum elatius*, the only species possessing both male and bisexual flowers sampled in the present study. When more than one species per genus were sampled, their P/O ratios differed considerably (in *Poa*, *Digitaria*, *Chloris*, *Echinochloa*, *Urochloa*).

## 4. Discussion

Except for some cleistogamous species, anemophily is the main pollination strategy in the family Poaceae. It is generally assumed that pollen production per anther (and flower) is controlled by genotype and is largely fixed [31]. Our study supports this hypothesis, as the pollen production per anther was one of the most stable parameters. Its coefficient of variation did not exceed 39% and, in most species, was less than 25%. Twelve of the species studied here have been previously studied by other authors (Table 3).

In some cases, the recorded differences in mean pollen production per flower between the present study and literature were insignificant, as the literature values were within the 95% confidence interval of the mean values inferred in our study (*Lolium perenne, Eleusine indica*, *Dactyloctenium aegyptium*, *Digitaria ciliaris,* two out of four literature records for *Dactylis glomerata*). In other instances (e.g., *Festuca pratensis*), literature data showed more than twofold difference from our observations (Table 3). Possible explanations of this disagreement may be the influence of local climatic conditions [61], different physiological status of a plant [62], differences in pollen production among varieties of a species [62] or the lack of a uniform pollen counting method. A case study of five species of *Bromus* revealed differences in anther length (and thus pollen production) between the lower and upper flowers of a spikelet [34]. In *B. inermis*, pollen production per anther was decreased to 77% by defoliation and increased to 132% in shoots that grew on thatching ant (*Formica obscuripes*) mounds [34]. Notably, in no case in the present study did we observe a twofold difference between Moscow and Ryazan samples of the same species. Similarly, a twofold level of difference was not found in the case study of McKone [34]. 

Possible differences in methods of counting pollen grains (and their preciseness) increases the significance of studies that include multiple grass species from different geographical regions and more than one sample per species. The present study provides a large dataset in this respect. Prieto-Baena et al. [26] presented highly important data on as many as 38 grass species from the city of Córdoba (Spain), but they did not compare samples of the same species from different localities and did not provide data on variations in pollen production per inflorescence, as we have done.

The P/O ratio is a conservative indicator of breeding systems and reflects their efficiency, i.e., the more efficient the transfer of pollen, the lower the P/O ratio [30,63]. This ratio usually increases in the direction of cleistogamy → obligate autogamy → facultative autogamy → facultative xenogamy → obligate xenogamy. In our study, the P/O ratios of obligate annuals varied from 318 to 1131 (Figure 6). These data suggest that the annual species studied here are most likely facultatively xenogamous. Indeed, in the dataset published by Cruden [30], who studied a range of angiosperm species, the mean P/O ratio of facultatively xenogamous species was 797 ± 88, whereas the mean values for facultative autogamy and obligate xenogamy were 169 ± 22 and 5859 ± 936, respectively. The differences in P/O values between species of the same genus revealed in the present study (*Chloris virgata* P/O = 379 and *C. barbata* P/O = 757; *Digitaria stricta* P/O = 318 and *D. ciliaris* P/O = 1132) may indicate different reproductive strategies of the species. The species of *Digitaria* studied here have similar pollen production per inflorescence, but differ significantly in the length of the anthers and the number of spikelets in the inflorescence. The difference in the P/O values of closely related species may be associated with a change in the reproductive strategy depending on environmental conditions [30]. The P/O values for *Digitaria sanguinalis* and *Echinocloa crus-gali* published by Cruden [30] were nearly two times higher than our results for the same species. Cruden [30] observed “dramatic intraspecific variations” of P/O for some eudicot species, which, in all cases, was related to intraspecific differences in flower size. This also can be attributed to different habitat conditions and the degree of disturbance therein [30]. We believe that this disagreement may relate to different sampling time (midgrowing season or its start/end, when the meteorological conditions can be unfavorable). In the obligately perennial species sampled here, P/O values varied from 3587 (*Tripogon jacquemontii*) to 32,970 (*Arrhenatherum elatius*), which, based on the figures provided by Cruden [30], suggest the occurrence of obligate xenogamy.

It is necessary to note that apart from the P/O ratio, another parameter could be useful in future studies of reproductive biology. Namely, it is reasonable to divide the pollen production by the total seed production. Detailed studies by Smith [64] based on native and exotic grasses of Pacific Northwest of the United States revealed great differences among grass species in terms of the percentage of seed set per flower. Among the species sampled here, the clonal perennial grass *Calamagrostis epigeios* is known to show a variable and generally low percentage of seed set per flower. The percentage of well-developed fruits ranged from c. 5% to c. 43% in populations examined in Germany [65,66]. Invasive North American populations of *C. epigeios* mostly do not produce seeds at all [67]. Notably, in the present study, pollen production per flower and the P/O ratio revealed *C. epigeios* to be among the lowest among perennial species sampled here. In contrast, the pollen production per inflorescence found in the present study for *C. epigeios* showed the highest figures among the grass species sampled here. We believe than when all data have been recalculated to Pollen/Seed ratios, the figures for *C. epigeios* should better fit the typical values of xenogamous perennial grasses. Therefore, we highlight a need for detailed studies of reproductive biology in grasses that include both pollen and seed production.

Apart from *Calamagrostis epigeios*, there is another perennial grass in our dataset with a remarkably low P/O ratio, *Tripogon jacquemontii*. We found no detailed observations regarding the reproductive biology of *T. jacquemontii*, but according to Thoiba and Pradeep [68], some Indian species of *Tripogon* are characterized by very low seed setting. In addition, *Tripogon jacquemontii* is remarkable in its ecology. This is apparently the only desiccation-tolerant grass sampled here [69].

Among the grass species studied here, the highest P/O ratio was found in *Arrhenatherum elatius*. This was the only species in our dataset that possesses male and bisexual flowers. The occurrence of male flowers should be viewed as an adaptation to cross-pollination, which likely explains the increased P/O ratio. Our P/O ratio count for *A. elatius* (32,970) was close to an earlier count (37,124) by Pohl (1937, cited after [63]). Based on data from Prieto-Baena et al. [27], another species of the genus, *Arrhenatherum album*, has a twofold higher P/O ratio than *A. elatius*.

According to Baumann et al. [70], all four species with the highest P/O ratios found in the present study (*Arrhenatherum elatius*, *Lolium perenne*, *Festuca pratensis*, *Alopecurus pratensis*) are likely self-incompatible. Data from Smith [64] allow us to add to this list *Dactylis glomerata*, *Bromus inermis* and *Elymus repens*. Therefore, all seven species with the highest P/O ratios (Figure 6) are self-incompatible. Why then does *Arrhenatherum elatius* still have a much higher P/O ratio than the six other species? An increased P/O ratio could be expected in a species with pronounced clonal growth (so that the distance to another genotype may be increased), but *A. elatius* is not capable of a clonal growth [71]. The high P/O ratio of *A. elatius* relative to other self-incompatible species sampled here can be plausibly explained by differences in the percentage of seed set. The average number of developed seeds per flower was estimated as 0.399 in *A. elatius*, a figure approaching those for *Dactylis glomerata* (0.417), *Bromus inermis* (0.375), and *Elymus repens* (0.248) [64]. However, all *Bromus*, *Elymus* and *Dactylis* flowers are female-fertile, whereas every second flower is male in *Arrhenatherum*. Therefore, relative to female-fertile flowers, the average number of developed seeds is close to 0.8 in *A. elatius*. This much higher seed set percentage is most likely facilitated by the high P/O ratio in *A. elatius*.

The example of *Arrhenatherum* shows that placing data in a broad context of reproductive biology is essential for interpreting pollen production in grasses, but currently available data in many instances remain insufficient for making such comparisons. As pointed out by Kellogg [9], given the ecological dominance of grasses, it is surprising how few studies have been undertaken on their breeding systems. Apparently, self-incompatibility is only rarely complete in grasses, though it is difficult to reject a hypothesis that the low levels of seed set still found in bagged plants of such species may be partly explained by apomixis [64]. Some grasses including *A. elatius* show certain genetically-determined infraspecific variations in their degree of self-incompatibility [64,71,72,73]. It will be interesting to learn whether genotypes that differ in degrees of self-incompatibility also differ in pollen production per anther. To our knowledge, such data are not available to date.

Pollen production per inflorescence is determined by a number of parameters, some of which are relatively stable (pollen production of an anther, number of stamens, number of flowers per spikelet), while others (number of spikelets per inflorescence) are more variable and seem to be, to a large extent, environmentally determined [27,28]. In some cases, lower production of pollen per anther (*Calamagrostis*) can be compensated for by larger numbers of spikelets per inflorescence, and vice versa, i.e., species with a lower number of flowers (*Lolium*) display very high pollen production per anther.

Our results support the hypothesis that pollen production per inflorescence differs significantly between annual and perennial species. The greater pollen production of perennial plants can be interpreted as a tendency to guarantee the cross-fertilization of species with self-incompatibility, which is typical for many perennial grasses [14,70] and plants from other angiosperm families [74,75,76]. Note that beyond the taxon sampling of the present study, there are self-incompatible species of annual grasses that predictably show high P/O ratio values. Most remarkably, the P/O ratio of the annual cereal crop rye (*Secale cereale*) is higher than those of any annual or perennial species sampled here, and indeed, higher than those of most other grasses examined to date [63]. P/O values greater than in *Secale* (c. 57,000) were only observed in *Leymus chinenis* (c. 80,000) and *Pariana stenolemma* (c. 300,000), with the latter being pollinated by beetles and flies [77]. One may speculate that the high P/O ratio of rye is in part due to domestication, because it maximizes the seed set under the conditions of self-incompatibility.

Among perennial plants, the lowest pollen production per inflorescence was noted for *Iseilema hackelii*, a species with complex spikelet clusters of dimorphic spikelets. The taxonomy of *Iseilema* is complicate. We applied the nomenclature adopted by Chorghe and Tiwari [78]. In our material, each cluster consisted of four well-developed male spikelets forming an involucre, two central sterile pedicelled male spikelets (with reduced anthers or without them at all) and a central sessile female spikelet. According to the literature, the female-fertile spikelets of *Iseilema* are bisexual, while pedicelled male spikelets are fertile [9,54,79,80]. Future studies should pay more attention to the presence or absence of stamens in female-fertile spikelets of *Iseilema*, as literature data in this respect are obviously incomplete. The pollen production of *Iseilema hackelii* is comparable with that of *Poa annua*, an annual or short-living perennial species [81] studied here using material from Russia. The pollen production of other studied perennial species from India, i.e., *Tripogon jacquemontii* and *Isachne elegans* (the latter species is only facultatively perennial [54]), was found to exceed those of obligatory annual species by 2–4 times, approaching those of perennial species from Russia (*Bromus*, *Elymus*, *Lolium*), which confirms the hypothesis of higher pollen production by perennial grasses. The two facultatively annual species studied here (*Poa annua* and *Isachne elegans*) surpassed all obligate annuals included in this study in terms of their pollen production per inflorescence.

Unlike Friedman and Harder [52], we found no strong correlation between pollen production and the size of pollen grains, but demonstrated a significant correlation between pollen production per anther and anther length. We believe that this provides an instrument for comparisons of the pollen production of grass species and estimates of pollen production in plant communities without requiring time-consuming calculations of pollen grains in anthers under a microscope. Instead, rough estimates may be made using much simpler measurements of anther length. Moreover, as anther length is so frequently used as a key characteristic in grass taxonomy, measurements are, in many cases, already available in literature. Our data support earlier studies that documented correlations between anther length and pollen production in particular taxa of grasses [36,37,39]. McKone [34] demonstrated that anther length is an excellent predictor of pollen production in *Bromus* (R^2^ = 0.96). The dataset generated in the present study provided a lower value of coefficient of determination (R^2^ =0.73). However, given the fact that we analyzed a set of distantly-related grass species belonging to different subfamilies and adapted to different ecological conditions, we consider the regression found in the present study to be conclusive and useful. The regression coefficients revealed for such varieties of wheat as “Gaby” (1050) and “Orca” (1072) by De Vries [39] were close to those found in the present study (1024). As McKone [34] provided coefficients of regression between anther length and volume of pollen instead of number of pollen grains, it is impossible to compare that result with ours.

The size of pollen grains is usually considered as a fairly stable feature. Our results show that the diameter of the pollen grains of *Bromus inermis*, *Dactylis glomerata*, *Lolium perenne*, *Phleum pratense* from different populations can differ considerably. We presume this to be related to different environmental conditions, primarily concerning mineral nutrition [82,83], even though the genetic heterogeneity of plant material cannot be excluded, especially in polyploid complexes. Such a wide range of variation should be taken into account when pollen size is used as an indicator of polyploidy or taxonomic character. In each specific case, such a relationship must be validated independently. For example, two samples of a taxonomically - and cytologically - homogenous species, i.e., *Aponogeton satarensis* (Aponogetonaceae), revealed ca. 1.4-times difference in mean pollen grain size [84].

## 5. Conclusions

Among the parameters that determine pollen production per inflorescence, pollen production per anther and the number of flowers per spikelet were found to be more stable than the number of spikelets per inflorescence, which is highly variable in nature. In some cases, lower production of pollen per anther can be compensated for by a larger number of spikelets per inflorescence, and vice versa, i.e., species with fewer flowers have very high pollen production per anther. The values of pollen production per flower in our study only partly agreed with previously published data. This disagreement may have been due to the influence of local climatic conditions, the different physiological status of various plants, differences in pollen production among genotypes or the lack of a uniform pollen counting method.

Our results support the hypotheses that pollen production per inflorescence differs significantly between annual and perennial species. The greater pollen production of perennial plants can be interpreted as a tendency to guarantee the cross-fertilization of species with self-incompatibility. In our study, the P/O values allowed us to suggest facultative xenogamy for all annual species and obligate xenogamy for most perennial species. However, self-incompatible annuals with high P/O values do exist among grasses not included in our dataset.

There is a correlation between pollen production per anther and anther length that allows rough comparisons to be made of the pollen production of grass species, and estimates to be made of pollen production in plant communities. This work provides new data that must be taken into account in phenological studies and aerobiological analyses.

Further research should aim to achieve a better understanding of the reproductive biology of individual grass species and biological interpretation of quantitative characteristics, such as pollen production and pollen/ovule ratio. We suggest the use of another parameter, namely, the pollen/seed ratio, that could be a finer indicator of the pollination strategies of various plant species. There is a wide field of research in refinement of biological interpretation of pollen production. For example, it is well-known that grass anthesis is normally restricted to a short period each day. In theory, the shorter the period of anthesis, the lower the P/O or P/S ratio, because a higher pollen concentration in air can be achieved. This hypothesis requires experimental testing. One would also expect a link to be observed between characteristics related to effective pollen dispersal (viability of pollen, its weight and size, inflorescene position above the substrate) and pollen production. Finally, much more detailed knowledge of patterns of incompatibility in various grass species is required, for example, to figure out to what extent plants with nonidentical genotypes existing in the same population are partly incompatible with each other [9].

## Figures and Tables

**Figure 1 plants-11-00285-f001:**
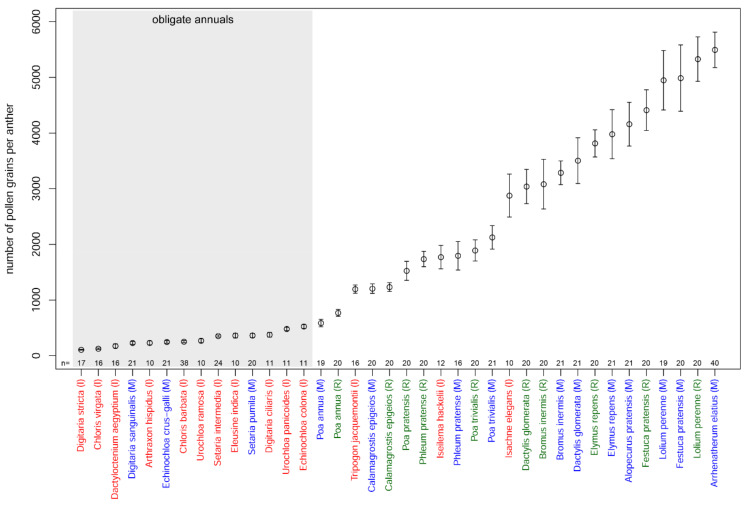
Pollen production per anther in studied samples; *n* = the number of anthers examined in each species. Mean values and their 95% confidence intervals are shown. Colors indicate samples from different regions: red (I), India; green (R), Ryazan; blue (M), Moscow.

**Figure 2 plants-11-00285-f002:**
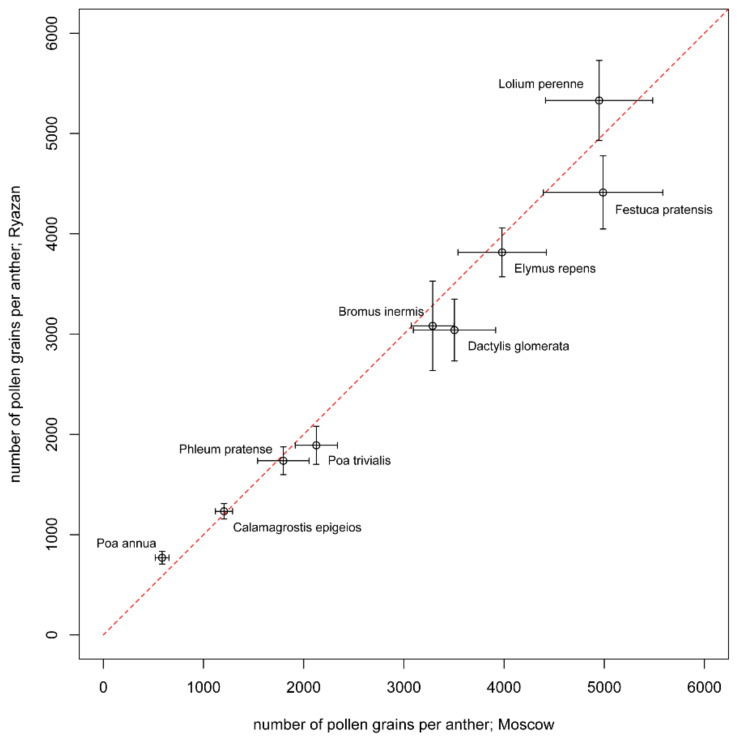
Comparison of samples from Moscow and Ryazan of studied species in terms of pollen production per anther. Mean values and their 95% confidence intervals are shown. The red line indicates equal values in both populations.

**Figure 3 plants-11-00285-f003:**
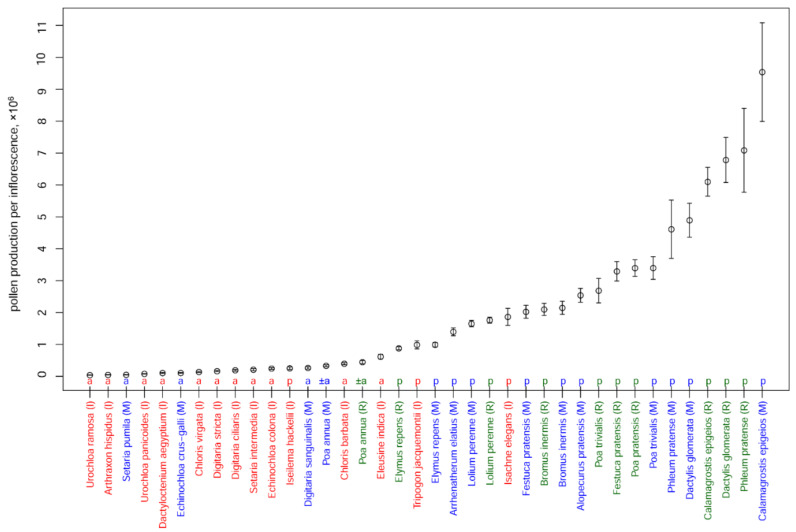
Pollen production per inflorescence in studied species; *n* = the number of inflorescences examined in each species. Mean values and their 95% confidence intervals are shown. a, obligate annual; p, perennial. Colors indicate samples from different regions: red (I), India; green (R), Ryazan; blue (M), Moscow.

**Figure 4 plants-11-00285-f004:**
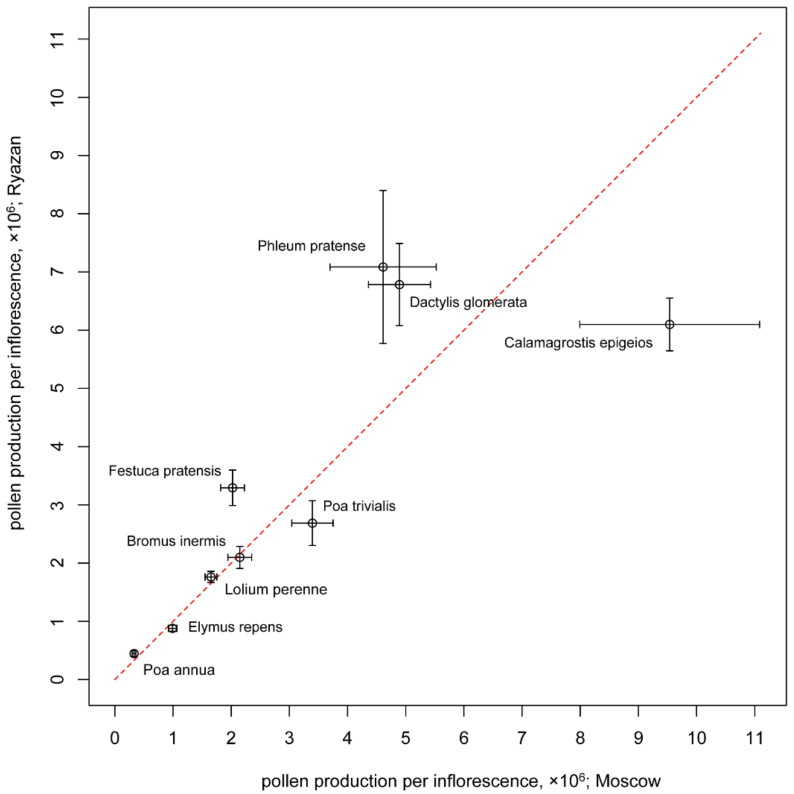
Comparison of samples from Moscow and Ryazan of studied species in terms of pollen production per inflorescence. Mean values and their 95% confidence intervals are shown. The red line indicates an equal value in both locations.

**Figure 5 plants-11-00285-f005:**
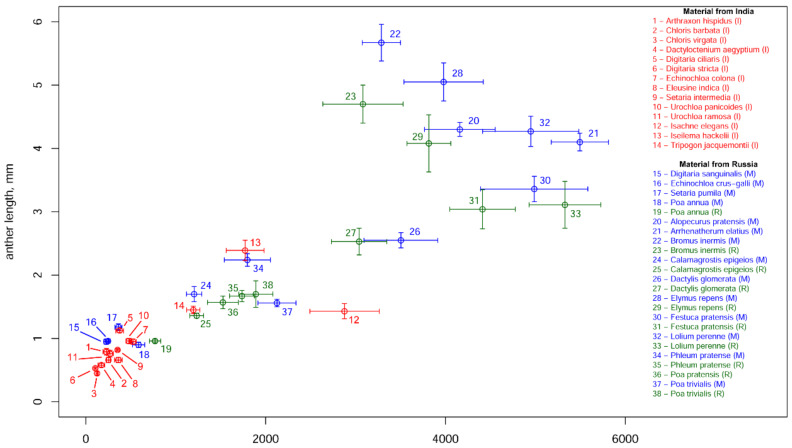
Scatter plot of pollen production per anther vs. anther length for studied species. Mean values and their 95% confidence intervals are shown. Pearson’s r between variables is 0.857 (*p*-value << 0.001); equation of linear regression is: pollen production = 1024.4 × length. I, India (red); R, Ryazan (green); M, Moscow (blue).

**Figure 6 plants-11-00285-f006:**
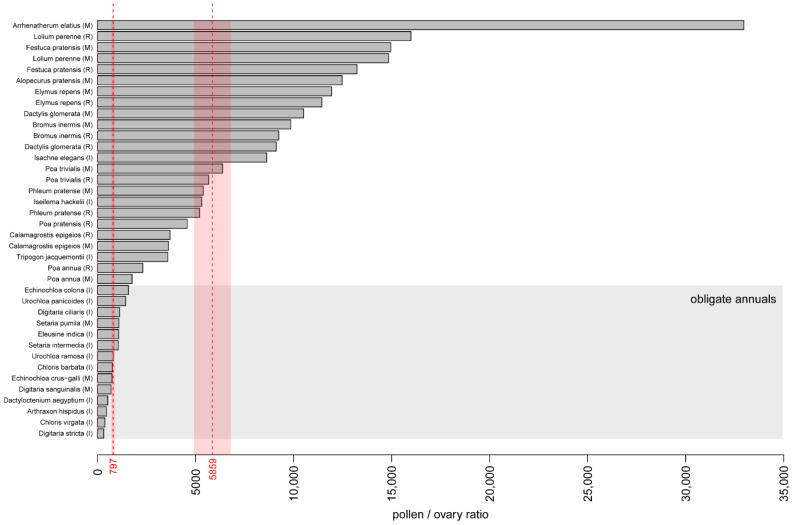
Pollen/ovule ratio of studied species. In the dataset of Cruden [30] who studied a range of angiosperm species, mean P/O ratio of facultatively xenogamous species was 797 ± 88 and that of obligate xenogamy was 5859 ± 936. These values are indicated here in red.

**Table 1 plants-11-00285-t001:** Material used in the present study.

Species	Voucher Information and GenBank Accession Numbers for ITS1-2 Sequences	Locality, Sampling Date	Life Form
*Alopecurus pratensis* L.	*Severova s.n.* (MW 1072500)	Lomonosov Moscow State University, Moscow, Russia, 28 May 2021	perennial
*Arrhenatherum elatius* (L.) P. Beauv. ex J. & C. Presl.	*Severova s.n.* (MW 1072498)	as above, 11 June 2021	perennial
*Arthraxon hispidus* (Thunb.) Makino	*Sokoloff et al. 127* (MW 0758387), OL752458	Shivaji University, Botanical Garden, Kolhapur, India, 31 August 2019	annual
*Bromus inermis* Leyss.	*Severova s.n.* (MW 1072504)	1. Lomonosov Moscow State University, Moscow, Russia, 11 June 2021 2. Sovetskyi rayion, Ryazan, Russia, 8 June 2021	perennial
*Calamagrostis epigeios* (L.) Roth	*Severova s.n.* (MW 1072496)	1. as above, 9 July 2021 2. Deulino, Ryazan’ oblast, Russia, 5 July 2020	perennial
*Chloris barbata* Sw.	*Sokoloff et al. 87* (MW 0758380); OL752448	Shivaji University, Botanical Garden, Kolhapur, India, 29 August 2019	annual
*Chloris virgata* Sw.	*Sokoloff et al. 93* (MW 0758383); OL752452	as above, 29 August 2019	annual
*Dactylis glomerata* L.	*Severova**s.n.* (MW 1072497)	1. Lomonosov Moscow State University, Moscow, Russia, 28 May 2021 2. Sovetskyi rayion, Ryazan, Russia, 9 June 2020	perennial
*Dactyloctenium aegyptium* (L.) Willd.	*Sokoloff et al. 88* (MW 0758384); OL752449	Shivaji University, Botanical Garden, Kolhapur, India, 29 August 2019	annual
*Digitaria ciliaris* (Retz.) Koeler	*Sokoloff et al. 92* (MW 0758386); OL752451	as above, 29 August 2019	annual
*Digitaria sanguinalis* (L.) Scop.	*Severova s.n.* (MW 1072506)	Lomonosov Moscow State University, Moscow, Russia, 4 October 2021	annual
*Digitaria stricta* Roth ex Roem. & Schult.	*Sokoloff et al. 98* (MW 0758385); OL752455 **	Shivaji University, Botanical Garden, Kolhapur, India, 29 August 2019	annual
*Echinochloa colona* (L.) Link	*Sokoloff et al. 86* (MW 0758390); OL752447	as above, 29 August 2019	annual
*Echinochloa crus-galli* (L.) P. Beauv.	*Severova s.n.* (MW 1072505)	Marfino, Moscow, Russia, 4 October 2021	annual
*Eleusine indica* (L.) Gaertn.	*Sokoloff et al. 97* (MW 0758388); OL752454	Shivaji University, Botanical Garden, Kolhapur, India, 29 August 2019	annual
*Elymus repens* (L.) Gould	*Severova s.n.* (MW 1072493)	1. Lomonosov Moscow State University, Moscow, Russia, 28 May 2021 2. Deulino, Ryazan oblast, Russia, 10 June 2020	perennial
*Festuca pratensis* Huds.	*Severova s.n.* (MW 1072495)	1. as above, 28 May 2021 2. Sovetskyi rayion, Ryazan, Russia, 10 June 2020	perennial
*Isachne elegans* Dalzell	*Sokoloff et al. 156* (MW 0758391); OL752459 **	Western Ghats, Maharashtra State, India, 2 September 2019	perennial
*Iseilema hackelii* Shrestha et Gandhi	*Sokoloff et al. 108* (MW 0758392); OL752456 *	Western Ghats, Maharashtra State, Satara District, India, 30 August 2019	perennial
*Lolium perenne* L.	*Severova s.n.* (MW 1072494)	1. Lomonosov Moscow State University, Moscow, Russia, 28 May 2021 2. Sovetskyi rayion, Ryazan, Russia, 12 June 2020	perennial
*Phleum pratense* L.	*Severova s.n.* (MW 1072502)	1. as above, 28 May 2021 2. as above, 12 June 2020	perennial
*Poa annua* L.	*Severova s.n.* (MW 1072501)	1. as above, 28 May 2021 2. as above, 6 June 2020	annual or short-living perennial
*Poa pratensis* L.	*Selezneva*, *Karaseva s.n.* (MW 1072499)	Sovetskyi rayion, Ryazan, Russia, 10 June 2020	perennial
*Poa trivialis* L.	*Severova s.n.* (MW)	1. Lomonosov Moscow State University, Moscow, Russia, 28 May 2021 2. Ryazan, Russia, 10 June 2020	perennial
*Setaria intermedia* Roem. & Schult.	*Sokoloff et al. 95* (MW 0758381); OL752453 *	Shivaji University, Botanical Garden, Kolhapur, India, 29 August 2019	annual
*Setaria pumila* (Poir.) Roem. & Schult.	*Severova s.n.* (MW 1072503)	Lomonosov Moscow State University, Moscow, Russia, 4 October 2021	annual
*Tripogon jacquemontii* Stapf	*Sokoloff et al. 111* (MW 0758389); OL752457	Western Ghats, Maharashtra State, Satara District, India, 30 August 2019	perennial
*Urochloa panicoides* P. Beauv.	*Sokoloff et al. 85* (MW 0758379); OL752446 *	Shivaji University, Botanical Garden, Kolhapur, India, 29 August 2019	annual
*Urochloa ramosa* (L.) T.Q. Nguyen	*Sokoloff et al. 90* (MW 0758382); OL752450	as above, 29 August 2019	annual

*, first ITS1-2 sequence for the species in the GenBank database; **, first sequence for the species in the GenBank database overall.

**Table 2 plants-11-00285-t002:** Pollen production, anther length and diameter of pollen grains of studied species. Sampling location: I, India, R, Ryazan, M, Moscow. Data are shown as mean value ± 95% confidence interval (coefficient of variation, %).

Species	Pollen Production per Anther	Number of Flowers per Spikelet	Number of Spikelets per Inflorescence	Pollen Production per Inflorescence, ×1000	P/O	Anther Length, mm	Diameter of Pollen Grain, μm
*Alopecurus pratensis* (M)	4160 ± 393 (20.7)	1 ± 0 (0)	203 ± 17.4 (17.7)	2539 ± 217 (27.5)	12,480 ± 1179 (20.7)	4.3 ± 0.11 (8.7)	30.08 ± 0.77 (6.03)
*Arrhenatherum elatius* (M)	5495 ± 318 (18.1)	2 ± 0 (0)	42 ± 7.3 (25.8)	1397 ± 122 (31.9)	32,970 ± 1908 (18.1) *	4.1 ± 0.14 (7.5)	35.44 ± 0.93 (6.79)
*Arthraxon hispidus* (I)	230 ± 37 (22.4)	1 ± 0 (0)	101 ± 10 (15)	46 ± 5 (27.2)	460 ± 74 (22.4)	0.79 ± 0.05 (8.6)	26.9 ± 2 (14.5)
*Bromus inermis* (M)	3287 ± 213 (14.2)	5.2 ± 0.3 (17.6)	42 ± 6.9 (35.2)	2148 ± 203 (42.7)	9861 ± 639 (14.2)	5.67 ± 0.29 (12.3)	38.84 ± 1.48 (8.35)
*Bromus inermis* (R)	3082 ± 446 (30.9)	6.6 ± 0.3 (17.4)	35 ± 5.3 (32.9)	2098 ± 189 (50.1)	9246 ± 1338 (30.9)	4.7 ± 0.3 (13.4)	47.52 ± 1.44 (6.49)
*Calamagrostis epigeios* (M)	1205 ± 85 (15.1)	1 ± 0 (0)	2638 ± 856 (42.2)	9538 ± 1546 (45.3)	3615 ± 255 (15.1)	1.7 ± 0.12 (18.2)	23.69 ± 1.06 (10.83)
*Calamagrostis epigeios* (R)	1233 ± 77 (13.4)	1 ± 0 (0)	1648 ± 151.5 (19.6)	6099 ± 452 (23.9)	3699 ± 231 (13.4)	1.36 ± 0.04 (7.6)	23.36 ± 0.5 (5.17)
*Chloris barbata* (I)	252 ± 20 (24.4)	1 ± 0 (0)	524 ± 75.4 (20.1)	397 ± 33 (32)	756 ± 60 (24.4)	0.66 ± 0.04 (16.1)	23.82 ± 1.1 (15.47)
*Chloris virgata* (I)	126 ± 12 (18)	1 ± 0 (0)	365 ± 82.8 (31.8)	138 ± 17 (37)	378 ± 36 (18)	0.45 ± 0.03 (13.5)	24.11 ± 1.15 (16.64)
*Dactylis glomerata* (M)	3505 ± 411 (25.8)	2.6 ± 0.3 (23.5)	180 ± 19.5 (23.1)	4893 ± 533 (43.1)	10,515 ± 1233 (25.8)	2.55 ± 0.12 (11.3)	30.56 ± 1.2 (8.45)
*Dactylis glomerata* (R)	3040 ± 308 (21.7)	3.5 ± 0.3 (27.8)	211 ± 36.4 (36.9)	6783 ± 706 (53.1)	9120 ± 924 (21.7)	2.53 ± 0.21 (17.9)	38.6 ± 2.19 (12.45)
*Dactyloctenium aegyptium* (I)	174 ± 37 (39.5)	2.9 ± 0.2 (10.9)	69 ± 7.5 (15.4)	103 ± 15 (44.4)	522 ± 111 (39.5)	0.58 ± 0.03 (14.7)	28.23 ± 0.96 (14.2)
*Digitaria ciliaris* (I)	377 ± 43 (17)	1 ± 0 (0)	169 ± 44.5 (36.9)	191 ± 28 (41.1)	1131 ± 129 (17)	1.12 ± 0.03 (4.1)	28.44 ± 1.27 (12.39)
*Digitaria sanguinalis* (M)	229 ± 28 (27.3)	1 ± 0 (0)	385 ± 69.8 (38.7)	264 ± 39 (48.6)	687 ± 84 (27.3)	0.95 ± 0.04 (8.5)	30.63 ± 1.65 (12.75)
*Digitaria stricta* (I)	106 ± 7 (13.4)	1 ± 0 (0)	511 ± 141.7 (38.8)	162 ± 22 (41.3)	318 ± 21 (13.4)	0.53 ± 0.01 (6.1)	26.53 ± 0.81 (11.04)
*Echinochloa colona* (I)	525 ± 36 (10.2)	1 ± 0 (0)	155 ± 26 (25.3)	244 ± 27 (27.4)	1575 ± 108 (10.2)	0.95 ± 0.04 (5.3)	35 ± 1.5 (7)
*Echinochloa crus-galli* (M)	247 ± 24 (20.5)	1 ± 0 (0)	142 ± 23 (50.6)	105 ± 15 (55.6)	741 ± 72 (20.5)	0.96 ± 0.03 (7.2)	28.57 ± 1.57 (16.52)
*Eleusine indica* (I)	362 ± 42 (16.3)	5.2 ± 0.5 (12.2)	109 ± 20.9 (26.7)	618 ± 75 (34.1)	1086 ± 126 (16.3)	0.66 ± 0.04 (9.6)	21.93 ± 0.87 (12.22)
*Elymus repens* (M)	3980 ± 441 (24.3)	4.3 ± 0.2 (22.4)	19 ± 1.5 (16.9)	991 ± 73 (37.9)	11,940 ± 1323 (24.3)	5.05 ± 0.3 (13.5)	32.87 ± 0.84 (5.91)
*Elymus repens* (R)	3815 ± 244 (13.6)	4.7 ± 0.2 (16)	16 ± 1.6 (20.3)	879 ± 51 (29.6)	11,445 ± 732 (13.6)	4.08 ± 0.45 (25)	33.23 ± 0.82 (5.98)
*Festuca pratensis* (M)	4988 ± 596 (25.5)	4.8 ± 0.4 (29.9)	28 ± 3.9 (29.7)	2024 ± 205 (51.3)	14,964 ± 1788 (25.5)	3.36 ± 0.2 (12.5)	31.04 ± 1.19 (9.72)
*Festuca pratensis* (R)	4412 ± 365 (17.7)	7.3 ± 0.4 (19.7)	34 ± 5.9 (37.3)	3293 ± 305 (47)	13,236 ± 1095 (17.7)	3.04 ± 0.31 (22.5)	31.21 ± 1.44 (9.83)
*Isachne elegans* (I)	2877 ± 386 (18.7)	2 ± 0 (0)	108 ± 26.7 (34.6)	1864 ± 266 (39.9)	8631 ± 1158 (18.7)	1.43 ± 0.12 (11.7)	22.29 ± 1.37 (16.55)
*Iseilema hackelii* (I)	1772 ± 211 (18.8)	1 ± 0 (0)	47 ± 17.6 (51.1)	253 ± 41 (67.4)	5316 ± 633 (18.8) **	2.39 ± 0.16 (11.3)	23.79 ± 1.06 (13.32)
*Lolium perenne* (M)	4949 ± 535 (22.4)	6.5 ± 0.3 (20.7)	17 ± 0.8 (13.9)	1653 ± 103 (34.1)	14,847 ± 1605 (22.4)	4.27 ± 0.24 (12.6)	34.01 ± 1.06 (6.68)
*Lolium perenne* (R)	5329 ± 399 (16)	6.5 ± 0.3 (19.3)	17 ± 0.9 (11.2)	1761 ± 96 (27.8)	15,987 ± 1197 (16)	3.11 ± 0.37 (25.4)	42.63 ± 2.56 (12.84)
*Phleum pratense* (M)	1797 ± 257 (26.9)	1 ± 0 (0)	855 ± 210.1 (52.5)	4612 ± 913 (60.6)	5391 ± 771 (26.9)	2.24 ± 0.1 (11.1)	37.46 ± 1.03 (8)
*Phleum pratense* (R)	1737 ± 138 (17)	1 ± 0 (0)	1360 ± 360.4 (56.6)	7086 ± 1315 (59.9)	5211 ± 414 (17)	1.67 ± 0.09 (14.8)	31.72 ± 1.32 (9.17)
*Poa annua* (M)	587 ± 68 (23.9)	4.2 ± 0.3 (23.4)	45 ± 6.5 (30.8)	330 ± 31 (47)	1761 ± 204 (23.9)	0.9 ± 0.04 (11.2)	25.23 ± 1.12 (10.97)
*Poa annua* (R)	770 ± 63 (17.6)	3.7 ± 0.3 (23.8)	34 ± 7.8 (43.6)	445 ± 51 (54.4)	2310 ± 189 (17.6)	0.96 ± 0.03 (7)	27.5 ± 0.75 (6.06)
*Poa pratensis* (R)	1525 ± 171 (24)	4 ± 0.2 (16.7)	187 ± 21.5 (24.6)	3394 ± 260 (39.1)	4575 ± 513 (24)	1.57 ± 0.1 (13.8)	32.25 ± 1.69 (10.87)
*Poa trivialis* (M)	2126 ± 211 (21.8)	2.1 ± 0.1 (20.1)	257 ± 47 (34.4)	3396 ± 355 (46.7)	6378 ± 633 (21.8)	1.56 ± 0.06 (9.6)	23.6 ± 1.83 (18.72)
*Poa trivialis* (R)	1891 ± 190 (21.4)	2.5 ± 0.2 (32.5)	189 ± 50.6 (57.1)	2686 ± 385 (73)	5673 ± 570 (21.4)	1.7 ± 0.21 (27)	23.24 ± 0.88 (9.23)
*Setaria intermedia* (I)	354 ± 18 (12.4)	1 ± 0 (0)	197 ± 75.6 (53.6)	209 ± 34 (55.4)	1062 ± 54 (12.4)	0.82 ± 0.02 (5.8)	23.29 ± 0.62 (11.2)
*Setaria pumila* (M)	363 ± 39 (22.9)	1 ± 0 (0)	45 ± 9.3 (44.5)	49 ± 8 (51.1)	1089 ± 117 (22.9)	1.18 ± 0.05 (9.6)	32.55 ± 0.91 (9.38)
*Tripogon jacquemontii* (I)	1196 ± 74 (11.6)	9 ± 2.3 (35.1)	31 ± 2.7 (10.5)	989 ± 129 (38.9)	3588 ± 222 (11.6)	1.45 ± 0.06 (7.8)	22.98 ± 1.11 (17.26)
*Urochloa panicoides* (I)	480 ± 33 (10.3)	1 ± 0 (0)	54 ± 5 (23.4)	77 ± 6 (25.7)	1440 ± 99 (10.3)	0.96 ± 0.04 (7.3)	24.5 ± 0.8 (5.6)
*Urochloa ramosa* (I)	269 ± 37 (19.1)	1 ± 0 (0)	49 ± 11 (36.7)	40 ± 7 (41.9)	807 ± 111 (19.1)	0.76 ± 0.05 (7.7)	27.5 ± 3.1 (18.6)

* P/O ratio of *Arrhenatherum elatius* was calculated for a spikelet consisting of one bisexual and one male flower. ** P/O value of *Iseilema hackelii* was calculated for a group of spikelets consisting of several male and one female spikelet.

**Table 3 plants-11-00285-t003:** Comparison of mean pollen production per flower according to the present study and literature. Values in bold are within the 95% confidence interval of the means obtained in our study.

	This Study: I, India; M, Moscow; R, Ryazan	Prieto-Baena et al. [27]	Aboulaich et al. [28]	Kaybeleva, Yudakova [32]	Tormo-Molina et al. [42]	Bai, Reddy [33]	Smart et al. [40]	Subba Reddi, Reddi [31]	Cruden [30]	McKone [34]
*Bromus inermis*	9861 (M) 9246 (R)									11,913
*Chloris barbata*	756 (I)					837		945		
*Dactylis glomerata*	10,515 (M) 9120 (R)	**10,425**	6429	**9240**	5431		3900			
*Dactyloctenium aegyptium*	522 (I)							**555**		
*Digitaria ciliaris*	1131 (I)							**1125**		
*Digitaria sanguinalis*	687 (I)								1234	
*Echinochloa crus-gali*	741 (I)								1267	
*Eleusine indica*	1086 (I)							810	**1111**	
*Elymus repens*	11,940 (M) 11,445 (R)	16,230								
*Festuca pratensis*	14,964 (M) 13,236 (R)			5151						
*Lolium perenne*	14,847 (M) 15,987 (R)						**16,200**			
*Poa annua*	1761 (M) 2310 (R)	1022	1216	5523						
*Poa pratensis*	4575 (R)			3735						
*Poa trivialis*	6378 (M) 5673 (R)		3386							

## Data Availability

Data is contained within the article.

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
