# Peer review of "Pollen Production of Selected Grass Species in Russia and India at the Levels of Anther, Flower and Inflorescence"

_plants, 2022, doi:10.3390/plants11030285_

Round 1
Reviewer 1 Report
The present paper refers to pollen production in grass species that occur in two regions with contrast climatic conditions (Central Russia and in India).
Such study are important for aerobiology and interpretation of airborne pollen data, the planning of therapy for grass pollen sensitive patients and environmental health. Pollen production data can be also relevant, e.g. for agronomy, forestry, the estimation of reproductive strategy of plant species (important in invasive plants), and interpretation of paleobotany data.
This study is good designed. The text is easy to follow. The data are accordingly visualized. The obtained data gives an instrument for quick comparisons of pollen production of grass species and estimates of pollen production in plant communities, which is of great importance for airborne allergens prediction.
I recommend the ms for publication after minor corrections.
P2 L 60 (p.1978), ????? It should be removed
P3 L 98 (p. 1869). Reference citation is enough
Fig 1. Please explain the colours used for the species names (red, blue, green)
Fig 3. Comment as for description in Fig 1 .
Reviewer 2 Report
Title
I suggest changing the title, as it should be concise and informative.
Abstract
- The Abstract should contain the research background, aim of the study, methodological information, the most important results, and a summary of specific conclusions
- 17 - 18. Correct this information: a large amount of pollen does not cause allergies (but allergens are a problem)
- 20-23 what information do the authors want to convey?
- 23-24. Specify the data
Keywords:
Eliminate phrases appearing in the title of the study.
Introduction
- 43-50.
Quote the relevant references and delete “e.g.” in “(e.g., [2,3])…” (e.g., [4])”
- give examples of “...ecological role”
Define “significance for human civilization”.
Describe “the interactions”.
- Provide information about allergens in the described group of plants (structural proteins, storage proteins, or plant stress proteins), e.g. allergen family, examples of allergens related to the research topic.
- 51 – 69.
- Give other examples of “…the most important cereal crop, wheat.’’
- The examples of the plant species should be closely related to the topic of the study (the authors mention wheat, Parianum, Olyra, Bambusa polymorpha, Chusquea abietifolia, Dendrocalamus stricta, Schizostachum zollingeri).
- What do the authors mean by “Pollen of Parianum has a well-sculpted exine [17](p.1978), 60 which agrees with the idea on the occurrence of entomophily in this taxon”?
- correct the spelling of “exine [17](p.1978)”
- I suggest using Latin names for the genus or species of insects
- which plant species were visited by the insect genera? This information should not be generalised.
This fragment requires re-edition of the text to refer to the conducted research.
- 70 – 76
- correct the spelling of “aerobiology[26,27,33–35].”
Provide also other references related to the research.
- 77- 89
Give the specific meaning of “for interpretation of allergy data, and therapy planning, but also for agronomy, environmental health, forestry etc.”
- Scanning or transmission electron microscopy is used for examining the exine sculpture - a bold statement; I propose introducing the correction “It is impossible to distinguish pollen of different genera using light microscopy”
- eliminate general information, provide data related to the research topic “..different grass species may have pollen of different allergenicity [37]”.
- 90 - 103
- eliminate mental shortcuts “It is well-known that grasses (and sedges) produce pollen in an unusual manner.”
- Complete the information and make the correction (l. 92-95) “The microspores/pollen per locule.”
- correct “<…>”, “[45](s. 1869).”
- 104 – 119
- provide more information instead of mental shortcuts (1) “we selected two regions with contrast climatic conditions”, (2) “with similar climatic conditions”.
- Give a rationale for the undertaken research
- provide detailed information on the aim of the study (complete the information step by step; the aim of the study must be clear and closely related to the conducted research).
- Eliminate mental shortcuts “most widespread grass species”
Material and methods
- 122 – 178
- the text requires the introduction of thematic paragraphs,
- provide data on the coordinates of the research area,
- complete the information on the statistical analyses
- 159 - 161
- complete the information on the number of inflorescences examined for each species, the number of spikelets per inflorescence, and the number of flowers per spike in the analysed species,
- supplement the quotation of references for the method, especially confirming the number of repetitions: “we selected three spikelets from each inflorescence.”
- 166 – 168
What staining was used?
- 172 – 174
- provide the number of measured anthers of each species,
- provide information on whether the anthers were collected from different flowers and different spikelets (how many anthers per one flower and per spikelet).
- 175 - 178
- Provide literature data confirming the sufficient number of repetitions for the measurements of the diameter of pollen grains (in most studies, the standard number of replicates for diameter measurements is 150-200 pollen grains).
- which diameter of pollen grains was measured?
- semi-fixed preparations should contain a glycerine solution (as plant material preparations in water dry quickly).
- it is redundant to provide the magnification of observations at the microscope
- specify the type of microscope, series and place of manufacture
- 147 – 150
Table 1
- correct the phrase “Voucher information)…”
- in the columns “Locality” and “life form”, I suggest logical correction to eliminate repetitions.
- complete the information in column 4
Results
Table 2
Make a graphical correction of the table, as there are many repetitions, e.g. (%).
All tables and figures should be understandable for the reader and include all necessary explanations, including those of statistical calculations and the number of measurements.
Discussion
- 299 – 302, eliminate citation of the same references sequentially: “plant [55], …. species [55] or the…”
- 311 - 332 “[29]”
- 356 – 364 “[26,27].”
- I suggest distinguishing similar subsections as in the “Results” section and describing the authors’ results comparing them with findings reported by other authors, and not the other way around.
- The possibilities of practical application of the results as well as trends for further research should be indicated.
Conclusions
The conclusions should be modified. In the present form, they constitute a summary of the research results.
References
- The References section contains many editorial errors; the section should be proofread in accordance with the guidelines for authors.
- All Latin species names should be written in italics. This must be corrected throughout the manuscript.

Reviewer 3 Report
The paper aims to analyze various pollen production characteristics in 15 species of grasses widespread in Central Russia and 14 species of grasses from India. The manuscript deserves publication as it is well planned, structured, well written and discussed. After careful review, I only detected several weaknesses:
Title
Please delete from the title “Towards quantification of roles of grass species in airborne pollen”, since the document does not include any study of the relationship between pollen production and the concentrations of grass pollen in the atmosphere.
Material and methods
A deep improvement of the methods about plants recollection should be included: dates, number of plants sampled of each specie for the production study…….
Results
A DNA protocol was included in the Material and methods chapter, but no information on this topic was included in the Results chapter. Please include.
Discussion
Some parts of the discussion chapter were a mix between the results not included in the Results chapter and their discussion. Please rearrange the text for lines 308 to 354 and figure 6. Figure 6 and lines 308 to 322 should be moved to the Results chapter.
Conclusions
The conclusions chapter should only support your own findings. Change the style, as in current form it looks like a repeat of the abstract.
Lines 402 to 405 are introductory. Delete them, as these sentences were stated earlier in the manuscript.
Lines 405 to 409 have a methodology character, delete them.
Round 2
Reviewer 2 Report
Dear Editor,
the manuscript in its current form may be processed for printing.
Authors should check the text for compliance with the authors' guidelines.
Kind regards.